# A Monitoring Method Based on Vegetation Abnormal Information Applied to the Case of Jizong Shed-Tunnel Landslide

**Qing Guo** [1,*] ⬤, **Lianzi Tong** [1,2] **and Hua Wang** [1,2]

1 Aerospace Information Research Institute, Chinese Academy of Sciences, Beijing 100094, China
2 School of Electronic, Electrical and Communication Engineering, University of Chinese Academy of Sciences, Beijing 100049, China
\* Correspondence: guoqing@aircas.ac.cn; Tel.: +86-010-82178961

**Abstract:** Landslides are one of the most dangerous natural disasters, which have affected national economic development and social stability. This paper proposes a method to indirectly monitor the deformation characteristics of landslides by extracting the abnormal vegetation information, especially for the inaccessible high-mountain landslides in southwestern China. This paper extracts the vegetation anomaly information in the Jizong Shed-Tunnel landslide which is located on the main traffic road to Tibet by the optical remote sensing Gaofen-1 (GF-1) data, and analyzes the temporal and spatial characteristics of the vegetation anomaly information through a time series. Then, we use the small baseline subsets interferometry synthetic aperture radar (SBAS-InSAR) technology to process Sentinel-1 data to obtain the time-series surface deformation information. Finally, we analyze and verify the results of the two methods. The results show that there is obvious vegetation coverage (VC) decline, with a maximum increasing percentage of 8.77% for the low and medium VC, and obvious surface deformation around the landslide, with the highest settlement rate of between 0 mm/year and 30 mm/year. Through the time-series analysis, we find that the change trends of the two methods are basically the same. This paper shows that the method of using abnormal vegetation information to monitor the Jizong Shed-Tunnel landslide has a certain degree of reliability and practicability. It can provide a new idea and effective supplement for landslide monitoring.

**Keywords:** landslide monitoring; Jizong Shed-Tunnel landslide; optical remote sensing; vegetation coverage; SBAS-InSAR; surface deformation

## 1. Introduction

Landslides are defined as the instability and destruction of rock, soil, or other artificial materials under the action of gravity. They have become one of the most dangerous natural disasters due to their suddenness and destructive power and being prone to secondary disasters, causing huge casualties and economic losses all over the world [1–4]. China is also one of the countries prone to landslide disasters. Especially in the southwest of China, due to the large undulation terrain, the loose soil structure, and heavy rainfall, landslide disasters are extremely prone to occurring [5–10]. The Jizong Shed-Tunnel landslide is located in the southwestern Yunnan Province. The area is rich in mountains and is located next to the G214 National Highway known as the "lifeline" of Sichuan and Tibet. Once a landslide occurs, it will block the normal passage of the G214 National Highway and cause serious casualties and economic losses. Therefore, the identification and continuous monitoring of landslides is an effective way to prevent and control landslide hazards.

Many scholars have done a lot of work on landslide monitoring. Common landslide monitoring methods include the global navigation satellite system (GNSS) methods and the interferometry synthetic aperture radar (InSAR) methods. The global positioning system (GPS), as one part of the GNSS system, is widely used. GPS monitoring has the advantages of high automation degree, high precision, no need to meet the visibility between monitoring

sites, and all-weather real-time monitoring [11]. In the era of single GPS, GPS methods already have many precise applications for monitoring landslides [12–14]. In recent years, with the development of GNSS technology, GNSS technology has more combined applications and is developing towards low costs. Peng et al. [15] have used the BeiDou Navigation Satellite System (BDS)/GPS single-point positioning method to effectively monitor sliding landslides. Notti et al. [16] have used the low-cost GPS to continuously monitor unstable slopes in northwestern Italy, which verifies the accuracy of the method. At the same time, Šegina et al. [17] developed a low-cost GNSS monitoring system for a deep-seated landslide in north-western Sloveni, further demonstrating the effectiveness of low-cost GNSS. However, GPS satellite signals are easily blocked in complex terrain and dense vegetation coverage areas, such as high-mountain areas, which affects the accuracy. Moreover, GPS collects data at points, so it is difficult to monitor landslides comprehensively [18,19].

InSAR technology has the characteristics of all-weather and all-day operation and can obtain large-area, long-term series of surface deformation information [20]. InSAR also can generate the regional digital elevation model (DEM) from paired radar images, which has particular significance for areas without terrain data and can provide basic data for landslide hazard assessments [21]. In 2016, He et al. [22] used the optical and InSAR technology to monitor loess landslides and accurately analyzed the change of landslide surface elevation. In 2019, Huang et al. [23] used the small baseline subsets (SBAS)-InSAR technology to monitor the deformation trend of Baige landslides, which proves the feasibility of SBAS-InSAR technology for landslide monitoring. In 2020, Jiang et al. [24] used the coherent scanner (CS)-InSAR Technology for monitoring potential landslides in western China. However, InSAR is susceptible to the phase delay of water vapor in mountainous areas. Simultaneously, the characteristics of InSAR technology slant-range imaging can easily cause registration errors and spatial baseline decoherence in areas with large terrain undulations [14,19]. In Yunnan Province, as the southwest region of China, the terrain is undulating. The high-mountain landslides in this area are mostly lush vegetation, and the high vegetation coverage can also affect the InSAR results.

Landslide creeping is a stage in the development process of landslide deformation which is slow for a long time and difficult to be detected at the beginning [25]. During the creep stage of the landslide, there will be sudden changes to the water level, the uplift of the soil slope, and the collapse or relaxation of the surrounding rock mass. This is reflected in the growth status of vegetation on the landslide. Of course, not all the landslides at the creeping stage show obvious characteristics of changes to surface vegetation, but some do exist (such as Baige landslide [26] and Su village landslide [27]). At present, many scholars use remote sensing images to study the vegetation in landslide areas. In 2012, Lu et al. [28] calculated the normalized difference vegetation index (NDVI) on Landsat5 TM data before and after the earthquake, and have studied the vegetation recovery status of the Maoxian landslide in Sichuan after the earthquake. In 2015, He and Zhang [29] used the NDVI value to set a threshold to identify landslides, extract landslide feature information, and perform area statistics. In 2020, Piroton et al. [30] used the NDVI difference values of the pre-landslide image from the post-landslide image as a complementary qualitative analysis for landslide monitoring. In 2022, Xun et al. [31] selected the NDVI as a feature describing the vegetation information for the extraction of potential landslides. However, most studies are aimed at monitoring the restoration of vegetation after landslides or as one of the characteristics of identifying landslides. The studies on the analysis and monitoring of landslide creep using vegetation cover changes are few currently.

Both GPS and InSAR methods have certain limitations in the high-mountain areas of southwest China. In 2020, Guo et al. [32] used Gaofen (GF) satellite data to explore the relationship between vegetation anomalies and landslides, taking the Xinmocun landslide as an example. Subsequently, by studying the Baige landslide, Guo et al. [33] believe that potential landslides in high-mountain areas can be preliminarily investigated economically and effectively through vegetation change. Therefore, in order to further verify the possibility of using vegetation changes to monitor landslides, this paper uses the method of

calculating the vegetation coverage to extract the abnormal vegetation information of the Yunnan Jizong Shed-Tunnel landslide from optical remote sensing Gaofen-1 (GF-1) data. Meanwhile, the SBAS-InSAR method, which more easily obtains more comprehensive monitoring the GPS method, is used to obtain the time-series surface deformation of the landslide area, to further support the indirect optical monitoring method. The two aspects are combined to analyze the temporal and spatial characteristics of landslide creep and verify the feasibility and effectiveness of the method in this paper. This study uses the abnormal vegetation information to indirectly monitor the Jizong Shed-Tunnel landslides in an effective and sequential manner and provides a new idea and monitoring technology for high-mountain landslides in the southwest region, which can effectively supplement the landslide monitoring methods.

## 2. Study Area and Data

### 2.1. Study Area

The Jizong Shed-Tunnel landslide is located in Ladong Mountain on the east bank of the Jinsha River, in Diqing Tibetan Autonomous Prefecture, Yunnan Province (Figure 1). The geographic coordinates of the center of the landslide source area are 99°23′43″E, 28°7′53″N. This is the active landslide that can be seen on the G214 National Highway. As one of the main highways of the Chinese transportation network, the G214 National Highway is the only main transportation road from Yushu area to Xining and Sichuan-Tibet. It is the economic line and lifeline of the surrounding areas. Therefore, maintaining the safety of G214 National Highway is an important task to guarantee the economic stability and social stability of the surrounding areas [34].

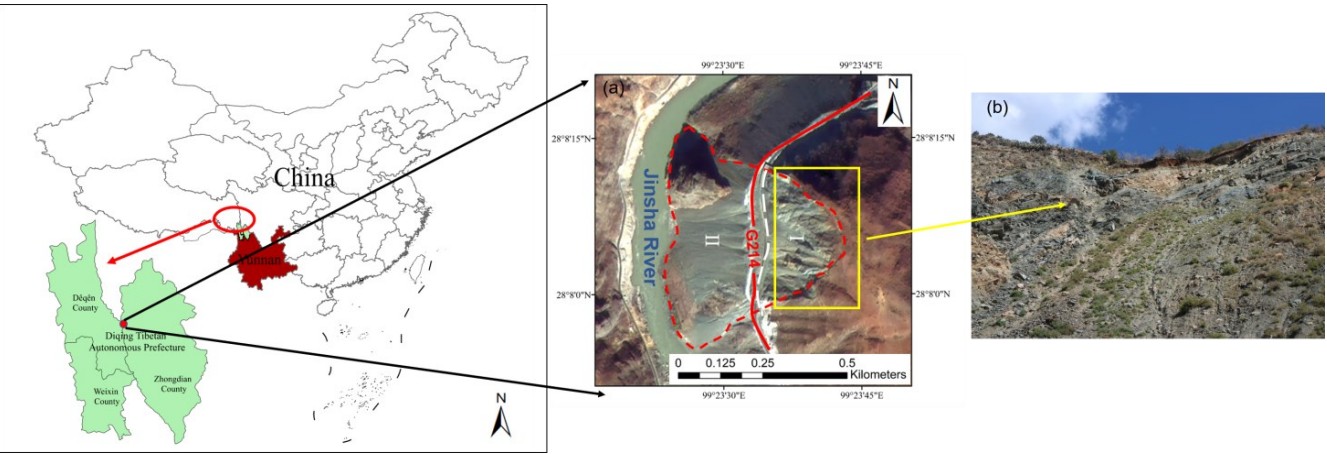

**Figure 1.** The location and Google Earth image of Jizong Shed-Tunnel landslide. (**a**) Optical GF-1 fused true color image; (**b**) Site investigation image.

The Jizong Shed-Tunnel landslide is near the normal fault, which is mainly composed of volcanic rock, slate, and limestone. The overall slope is between 30° and 45°. Moreover, there is abundant precipitation in the study area. Due to road construction and precipitation, the Jizong Shed-Tunnel landslide experienced a large slide in 2015, then the landslide was in a slow creep stage. The landslide moves along the slope layer from the top of the hillside and slides to the Jinsha River for accumulation, which can be divided into the source area (Figure 1a(I)) and the accumulation area (Figure 1a(II)) of the landslide. The creep of the landslide is mainly caused by the slight deformation and cracks at the rear edge of the upper landslide area. Therefore, the main analysis area is located in the upper landslide area.

### 2.2. Data

This paper selects GF-1 optical images for vegetation anomaly information extraction, Sentinel-1 A satellite radar images for deformation monitoring, and shuttle radar topogra-

phy mission digital elevation model (SRTM DEM) data as the auxiliary external DEM to eliminate the influence of terrain factors on the deformation monitoring.

### 2.2.1. GF-1 Optical Image

The GF-1 satellite is equipped with a panchromatic/multispectral PMS camera. The PMS camera can acquire panchromatic (PAN) images with a resolution of 2 m and multispectral (MS) images with a resolution of 8 m (blue, green, red, and near-infrared 4 bands), while the imaging width is 60 km. Thus, the GF-1 satellite provides reliable data formations for Earth observation.

Because the high cloud cover in the Jizong Shed-Tunnel landslide in the summer and the growth of vegetation in spring is easily affected, this paper selects eight GF-1 datapoints around November from 2013 to 2020 to analyze the abnormal vegetation information in order to avoid the influence of the season on the growth of plants. The image data are shown in Table 1.

**Table 1.** GF-1 image data.

| Number | Image Time | Number | Image Time | Number | Image Time |
|--------|------------|--------|------------|--------|------------|
| 1 | 5 November 2013 | 4 | 22 December 2016 | 7 | 24 November 2019 |
| 2 | 8 November 2014 | 5 | 11 November 2017 | 8 | 30 November 2020 |
| 3 | 16 November 2015 | 6 | 19 November 2018 | | |

### 2.2.2. Sentinel-1 A Radar Image

The Sentinel-1 satellite which carries a C-band synthetic aperture radar is composed of Sentinel-1 A and Sentinel-1 B. It provides reliable and repeated wide-area monitoring all-day and through all weather, so it can be used to obtain surface deformation and monitor large-scale resources. It has four working modes: stripmap (SM), interferometric wide swath mode (IW), extra wide swath mode (EW), and wave mode (WM). The Sentinel-1 satellite has an ultra-high radiation resolution and excellent coverage performance and revisit performance, which meets the research requirements of this paper.

Since the time interval of the Sentinel-1 satellite data in the study area from 2014 to 2016 was too large, and this paper mainly studies the subsequent landslide creep stage after 2015, in order to ensure the correlation between the InSAR data, this paper selects 57 scenes of the Sentinel-1 A satellite's single look complex (SLC) data with the IW working mode from 2017 to 2020 with the largest time interval of 24 days. All images are from the ascending orbit data with the same orbit path number 99 and frame number 1270. The image data are shown in Table 2.

**Table 2.** Sentinel-1 A image data.

| Number | Image Time | Polarization | Number | Image Time | Polarization |
|--------|------------|--------------|--------|------------|--------------|
| 1 | 18 March 2017 | VV | 30 | 12 February 2019 | VV |
| 2 | 30 March 2017 | VV | 31 | 8 March 2019 | VV |
| 3 | 23 April 2017 | VV | 32 | 1 April 2019 | VV |
| 4 | 17 May 2017 | VV | 33 | 25 April 2019 | VV |
| 5 | 10 June 2017 | VV | 34 | 19 May 2019 | VV |
| 6 | 4 July 2017 | VV | 35 | 12 June 2019 | VV |
| 7 | 9 August 2017 | VV | 36 | 6 July 2019 | VV |
| 8 | 2 September 2017 | VV | 37 | 30 July 2019 | VV |
| 9 | 26 September 2017 | VV | 38 | 23 August 2019 | VV |
| 10 | 20 October 2017 | VV | 39 | 16 September 2019 | VV |
| 11 | 13 November 2017 | VV | 40 | 10 October 2019 | VV |

**Table 2.** *Cont.*

| Number | Image Time | Polarization | Number | Image Time | Polarization |
|--------|------------|--------------|--------|------------|--------------|
| 12 | 7 December 2017 | VV | 41 | 3 November 2019 | VV |
| 13 | 31 December 2017 | VV | 42 | 27 November 2019 | VV |
| 14 | 24 January 2018 | VV | 43 | 21 December 2019 | VV |
| 15 | 17 February 2018 | VV | 44 | 14 January2020 | VV |
| 16 | 13 March 2018 | VV | 45 | 7 February2020 | VV |
| 17 | 6 April 2018 | VV | 46 | 2 March 2020 | VV |
| 18 | 30 April 2018 | VV | 47 | 26 March 2020 | VV |
| 19 | 24 May 2018 | VV | 48 | 19 April 2020 | VV |
| 20 | 17 June 2018 | VV | 49 | 13 May2020 | VV |
| 21 | 11 July 2018 | VV | 50 | 6 June 2020 | VV |
| 22 | 4 August 2018 | VV | 51 | 30 June 2020 | VV |
| 23 | 28 August 2018 | VV | 52 | 24 July 2020 | VV |
| 24 | 21 September 2018 | VV | 53 | 17 August 2020 | VV |
| 25 | 15 October 2018 | VV | 54 | 10 September 2020 | VV |
| 26 | 8 November 2018 | VV | 55 | 4 October 2020 | VV |
| 27 | 2 December 2018 | VV | 56 | 28 October 2020 | VV |
| 28 | 26 December 2018 | VV | 57 | 21 Nevember 2020 | VV |
| 29 | 19 January 2019 | VV | | | |

### 2.2.3. SRTM DEM Data

SRTM DEM data are synthesized in the United States using the SRTM system to obtain the radar image data from 60 degrees north latitude to 60 degrees south latitude.

SRTM DEM data can be divided into SRTM 1 (resolution of 30 m) and SRTM 3 (resolution of 90 m) data. This article uses SRTM 1 data with a latitude and longitude span of $1° \times 1°$. In order to cover the study area, we selected the SRTM N28E099 and N27E099 data.

## 3. Methods

### 3.1. Image Pre-Processing Method

Image fusion can complement the feature attributes of different data and make up for the incompleteness and uncertainty caused by single pieces of information, which is beneficial to the target recognition, analysis, and application of remote sensing [35].This paper adopts and compares the five fusion methods which are generally accepted by the public: nearest neighbor diffusion (NND) [36], principal component analysis (PCA) [37], Gram-Schmidt (GS) [38], high pass filter (HPF) [39], and Pansharpening fusion [40], and analyzes the fusion method suitable for the Jizong Shed-Tunnel landslide area. This enables MS images to improve the spatial resolution while ensuring the spectral information is unchanged as much as possible, so as to enhance the visual interpretation effect of the Jizong Shed-Tunnel landslide and improve the ability to detect vegetation growth and change information.

### 3.2. Vegetation Abnormal Information Extraction Method

In ideal theory, since this paper obtains optical images at nearly the same time every year for the same study area, the vegetation information is basically unchanged. Therefore, when there is obvious vegetation change information, it means that the vegetation in this area is abnormal. We use the vegetation coverage (VC) to obtain the information about vegetation abnormalities caused by the landslide creep. The VC is an important parameter to describe the ground vegetation cover. The commonly used remote sensing calculation method for VC is to estimate it based on the vegetation index. This paper adopts the pixel dichotomy model proposed by Li [41] and uses the vegetation index to estimate the VC. Since the vegetation in the study area is susceptible to the influence of the bare soil background, NDVI which is commonly used is difficult to apply to this area, while other vegetation indexes such as the green normalized difference index (GNDVI) can avoid this phenomenon. Therefore, this paper chooses the GNDVI to conduct the experiment and

finds that GNDVI can indeed be well applied in the study area. So, we finally chose GNDVI as the vegetation index of the study area. GNDVI, as the vegetation index that extracts the vegetation information and accurately reflects the vegetation coverage through the ratio processing, can eliminate the errors of the altitude of the Sun, the atmospheric attenuation, and the terrain changes. The formula is shown in Formula (1):

$$GNDVI = \frac{\rho_{NIR} - \rho_G}{\rho_{NIR} + \rho_G} \qquad (1)$$

Herein, $\rho_{NIR}$ is the reflectivity in the near-infrared band, and $\rho_G$ is the reflectivity in the green band.

The specific formula of the pixel dichotomy model to calculate VC is shown in (2):

$$VC = \begin{cases} 0 & , \ GNDVI \leq GNDVI_{soil} \\ \frac{GNDVI - GNDVI_{soil}}{GNDVI_{veg} - GNDVI_{soil}}, & GNDVI_{soil} \leq GNDVI \leq GNDVI_{veg} \\ 1 & , \ GNDVI \leq GNDVI_{soil} \end{cases} \qquad (2)$$

Herein, VC is the value of vegetation coverage. $GNDVI_{soil}$ is the value of GNDVI in the bare soil or areas without vegetation cover, and $GNDVI_{veg}$ is the value of GNDVI in areas completely covered by plants, which are pure vegetation pixels. Low VC values near 0 represent completely barren surfaces (rock or soil) or no vegetation-covered areas, while high VC values near 1 represent luxuriant vegetation.

When using the pixel dichotomy model to calculate the VC, the most important thing is to obtain the values of $GNDVI_{soil}$ and $GNDVI_{veg}$. In the actual situation, according to the definition of the parameters, these two values will change with time and space [42]. At present, we mainly count the value of GNDVI from the remote sensing images and set the confidence interval according to the cumulative percentage to define the value of $GNDVI_{soil}$ and $GNDVI_{veg}$. Because there are many remote sensing images in this article, in order to avoid the result analysis error caused by different confidence interval selections, this article makes multiple adjustments and calculations to obtain a unified confidence interval. The VC calculation flow chart is in Figure 2, and the specific operation steps are as follows:

(1) Calculate the initial $GNDVI_{soil}$ and $GNDVI_{veg}$ values of each time phase. We count the cumulative percentage of each GNDVI value in the image at first and select the initial confidence interval based on empirical values. We first use the 5–95% confidence interval [43] as the initial value to try. Then, we calculate the initial $GNDVI_{soil}$ and $GNDVI_{veg}$ based on the left and right boundaries of the confidence interval;

(2) Adjust the confidence interval for each time phase. We calculate the VC by using the values of $GNDVI_{soil}$ and $GNDVI_{veg}$$GNDVI_{veg}$ determined in step (1) and enhance results through the pseudo color density segmentation to visually judge the agreement degree of the bare land and vegetation area between in the VC map and in the original image. If not, repeat the above steps to redefine the confidence interval and perform the calculation again until the obtained result is the optimal fit;

(3) Determine the final uniform confidence interval. In order to unify the thresholds of each time phase and obtain the consistent and best-fitting VC as much as possible, we comprehensively consider the confidence intervals of each phase and unify them to obtain the final confidence interval consistent with each time phase.

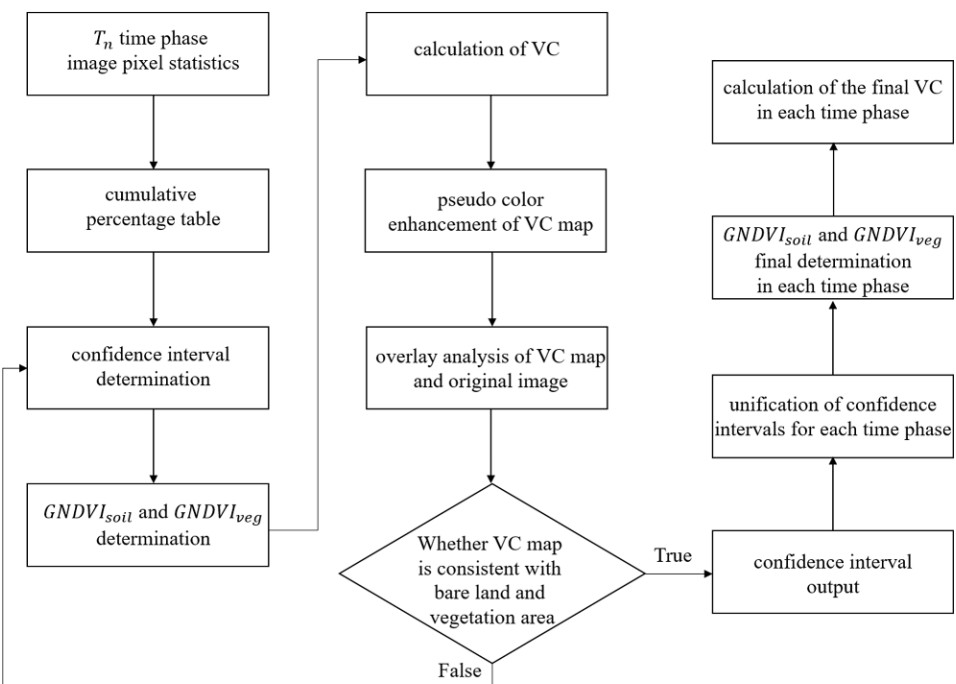

**Figure 2.** The VC calculation flow chart.

### 3.3. SBAS-InSAR Method

The traditional differential (D)-InSAR method is affected by factors such as the baseline length, atmospheric propagation delay, and spatial resolution, so it cannot be monitored well in most areas [44,45]. In recent years, researchers have successively proposed time-series InSAR methods such as the permanent scatterer (PS)-InSAR technology [46,47] and SBAS-InSAR technology [48] with the development of InSAR technology. The coherence requirements of PS points are higher than those obtained by SBAS, so PS-InSAR technology is often used for deformation monitoring in urban areas, and the SBAS-InSAR method is more effective than the PS-InSAR method in monitoring landslides in mountainous areas. Meanwhile, SBAS-InSAR is based on multi-master images, which overcomes the poor coherence shortcoming of some interferograms caused by using only one master image. Moreover, in SBAS-InSAR where only interferograms with small baselines are selected for the time-series analysis, the deformation results are denser and more reliable. At the same time, SBAS-InSAR method can still monitor the deformation rate in the vegetation coverage area [49]. The study area of this paper is with the vegetation coverage and in the mountainous area, so this paper chooses the SBAS-InSAR method to study the surface deformation of the Jizong Shed-Tunnel landslide.

The SBAS-InSAR method is a time-series InSAR method proposed by Berardino et al. [48]. This method mainly uses multiple synthetic aperture radar (SAR) images as the main image and forms different short baseline subsets according to the principle of short baseline interference to generate differential interference images. Then, based on the least square rule, the average surface deformation rate of the study area and the time series of the surface deformation are obtained by using the singular value decomposition (SVD) [48,50].

The basic principles of the SBAS-InSAR method are as follows:

(a) When there are $N$ scenes of SAR images in the study area, each SAR image will be differentially interfered with by at least another $N-1$ scenes image to form an interference image pair. Finally, $M$ interference image pairs will be obtained. Meanwhile, the

image with most interference pairs is chosen as the main image, and the rest of the images are the slave images. The value range of $M$ is shown in Formula (3) [51]:

$$N/2 \leq M \leq (N(N-1))/2 \tag{3}$$

(b)  Differential interferogram is collected from $M$ interferometric pairs by using the InSAR phase deformation extraction method. The final interferogram is obtained through the phase filtering and unwrapping. The interference phase of the $j$-th interferogram $\varphi_j$ can be expressed as Formula (4):

$$\Delta\varphi_j\,(x,y) = \varphi_B\,(x,y) - \varphi_A\,(x,y) \approx 4\pi/\lambda\,[d(t_B,x,y) - d(t_A,x,y)] \tag{4}$$

Herein, $t_A$ is the acquisition time of the main image, $t_B$ is the acquisition time of the slave image, $\lambda$ is the central wavelength, and $x$, $y$ are the azimuth and distance coordinates of the image, respectively, $\varphi_A$, $\varphi_B$ are the interference phase of the main image and the slave image, respectively.

The interferograms after the phase filtering and unwrapping are arranged in the time order of the image, and then, the vector phase of the interferogram can be directly expressed in the form of matrix. In the matrix, each row corresponds to a differential phase interferogram and each column corresponds to the SAR images at different times. The column of the main image and the slave image in the matrix is $\pm 1$, and the remaining columns are all 0, as shown in Formula (5):

$$G\varphi = \Delta\varphi \tag{5}$$

Herein, $G$ is an $M \times N$ matrix, expressed as: $G = \begin{bmatrix} 0 & -1 & 0 & 1 & \cdots & 0 & 0 & 0 \\ -1 & 0 & 1 & 0 & \cdots & 0 & 0 & 0 \\ \vdots & \vdots & \vdots & \vdots & \vdots & \vdots & \vdots & \vdots \\ 0 & 0 & 0 & 0 & \cdots & -1 & 0 & 1 \end{bmatrix}$,

$\varphi$ is the interference phase.

(c)  The $G$ matrix is solved by using the SVD method through the least square rule, as shown in Formula (6):

$$G = USV^T \tag{6}$$

Herein, $U$ is the orthogonal matrix, $S$ is the diagonal matrix, $V^T$ is the average phase rate. The solving equation of $V^T$ is as shown in Formula (7):

$$V^T = \left[ V_1 = \frac{\varphi_1}{t_1 - t_0}, \cdots V_N = \frac{\varphi_N - \varphi_{N-1}}{t_N - t_{N-1}} \right] \tag{7}$$

(d)  Through the above steps, the optimal solution of the velocity vector can be obtained, and thus, the surface deformation information can be obtained. The surface deformation information still has the atmospheric delay and other errors, so they need to be filtered to obtain the final accurate surface deformation information [8].

## 4. Results

### 4.1. Evaluation of Image Preprocessing Results

In this paper, we first performed radiometric correction to convert the DN value into the surface reflectance in PAN and MS images and performed geometric correction to eliminate geometric distortion. Then, we performed five fusion methods for PAN image and MS image: NND, GS, PCA, HPF, and Pansharpening fusion. This paper uses the ENVI tool to perform NND, GS, and PCA fusion, uses the ERDAS IMAGINE 9.2 tool to perform HPF fusion, and uses the PIE tool to perform Pansharpening fusion. The five fusion methods have achieved good visual effects. At the same time, the fidelity of the spectral performance is an important index for evaluating image fusion applications. Therefore,

we focus on the comparison and analysis of these five fusion methods from the aspect of spectral performance. After the comprehensive evaluation of these indicators, the fusion method with the best overall performance will be adopted.

### 4.1.1. The Spectral Curve of Image Fusion Features

Since we need to mainly extract the abnormal vegetation information in optical images for our landslide research, it is important to distinguish the vegetation from the bare land. In this paper, we compare and analyze the five fusion methods by viewing the shape and range of the spectral reflectance curves of vegetation and bare land, as shown in Figures 3 and 4.

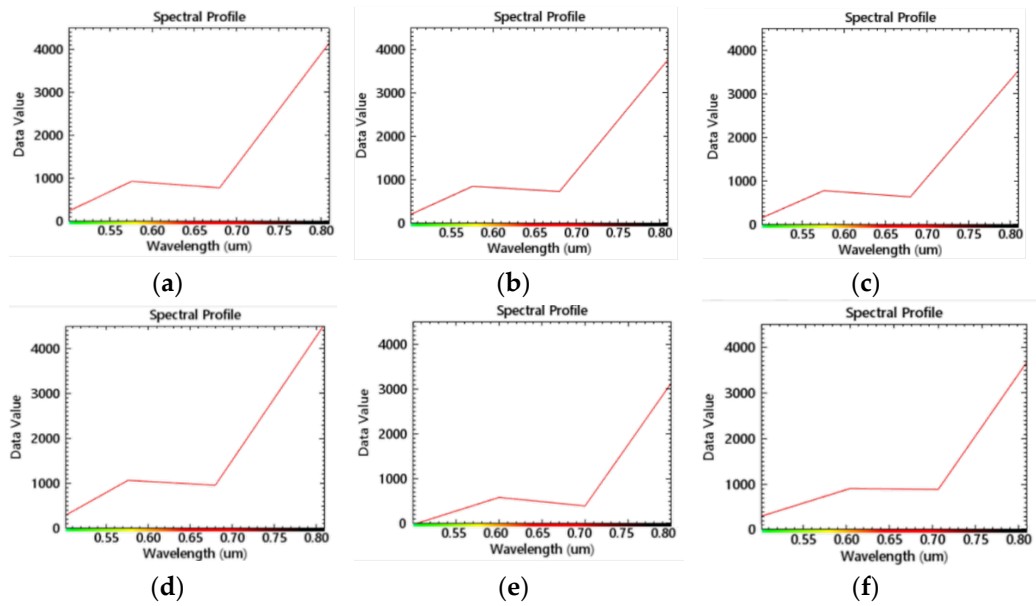

**Figure 3.** The vegetation spectrum curves of different fusion images. (**a**) MS image; (**b**) GS fusion image; (**c**) PCA fusion image; (**d**) NND fusion image; (**e**) Pansharpening fusion image; (**f**) HPF fusion image.

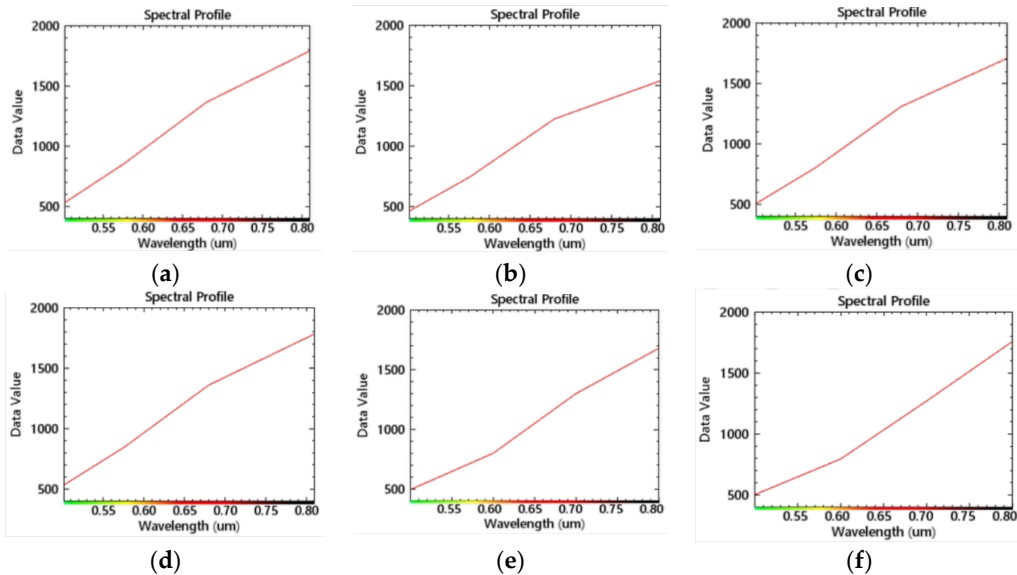

**Figure 4.** The bare ground spectrum curves of different fusion images. (**a**) MS image; (**b**) GS fusion image; (**c**) PCA fusion image; (**d**) NND fusion image; (**e**) Pansharpening fusion image; (**f**) HPF fusion image.

It can be seen from Figure 3 that in the vegetation reflectance spectrum curve, the vegetation spectrum curves of the five fusion images are consistent with the curve trend

of the MS image. In terms of the reflectance spectrum curve range, GS, PCA, HPF, and Pansharpening fusion images are all lower in the near-infrared, and only the NND fusion images tend to have the same range as a whole, which preserves the good spectrum information of NND.

For Figure 4, about the reflectance spectrum curve on the bare ground, the curve trend of the HPF fusion image is obviously inconsistent with that of MS image, and the trends of the other fusion images are relatively consistent. In terms of the reflectance spectral curve range, the NND, PCA, and Pansharpening fusion image and MS image tend to be the same, but the overall reflectance of the GS fusion image was relatively lower.

According to the spectral reflectance curves of bare land and vegetation, the overall quality of the five fusion methods is NND fusion method > PCA fusion method > Pansharpening > GS fusion method > HPF fusion method.

### 4.1.2. GNDVI Results

This paper needs to calculate the VC in the subsequent extraction of abormal vegetation information. The basis of calculating the VC is to calculate the GNDVI. Therefore, this paper chooses the GNDVI as the index of image fusion quality evaluation. The GNDVI results calculated from the above five fusion images are shown in Table 3.

**Table 3.** Comparison of GNDVI statistical results of different fusion images.

| GNDVI | MS | NND | GS | PCA | HPF | Pansharpening |
|---|---|---|---|---|---|---|
| Mean | 0.27 | 0.27 | 0.26 | 0.28 | 0.28 | 0.28 |
| Max | 0.68 | 0.69 | 0.75 | 0.72 | 0.74 | 0.74 |
| Min | 0.03 | 0.03 | $-0.39$ | $-0.17$ | $-0.22$ | $-0.27$ |

According to the GNDVI data in Table 3, it can be seen that the overall GNDVI value of the fused image is not much different from the MS image. Among them, the GNDVI value of the NND fusion image is the closest to the MS image, and the fidelity of the NND spectral information is the best.

After comprehensive evaluation of the above three different indicators, the NND fusion image has better performance in the details of the visual effect and the fidelity of the spectral curve. At the same time, the obtained GNDVI value of the NND fusion is also the closest to the MS image, which proves that it is reasonable and effective to use NND fusion images for subsequent vegetation monitoring.

### 4.2. Vegetation Abnormal Information Extraction Results Based on GF-1 Images

This paper determines 6% and 94% as the confidence interval of GNDVI according to the above method of determining the confidence interval. Finally, the VC is calculated and displayed in pseudo color. The VC results between 2013 and 2020 are shown in Figure 5.

During the creeping stage of the landslide, the slight deformation of the slope and the shear failure will cause cracks in the trailing edge. Based on this phenomenon, this paper mainly analyzes the vegetation cover change at the trailing edge of the landslide. Therefore, this paper mainly analyzes the change in VC at the trailing edge of the landslide. However, there are the landslide body and landslide trailing edge in the pseudo-color map of VC in Figure 5. In order to make an accurate analysis, this paper uses a black line to distinguish. On the left side of the black line, it is the landslide body. On the right side of the black line, it is the vegetation on the trailing edge of the landslide and the hillside.

According to the pseudo-color VC map of the landslide trailing edge shown in Figure 5, the overall VC shows a downward trend from 2013 to 2014 due to the large landslide in 2015. Compared with 2015, the VC at the trailing edge of the landslide has a certain recovery in 2016. The overall VCs change a little from 2017 to 2020, but from the perspective of subdivision areas, the changes in VC can be observed between the red box area and the green box area in Figure 5.

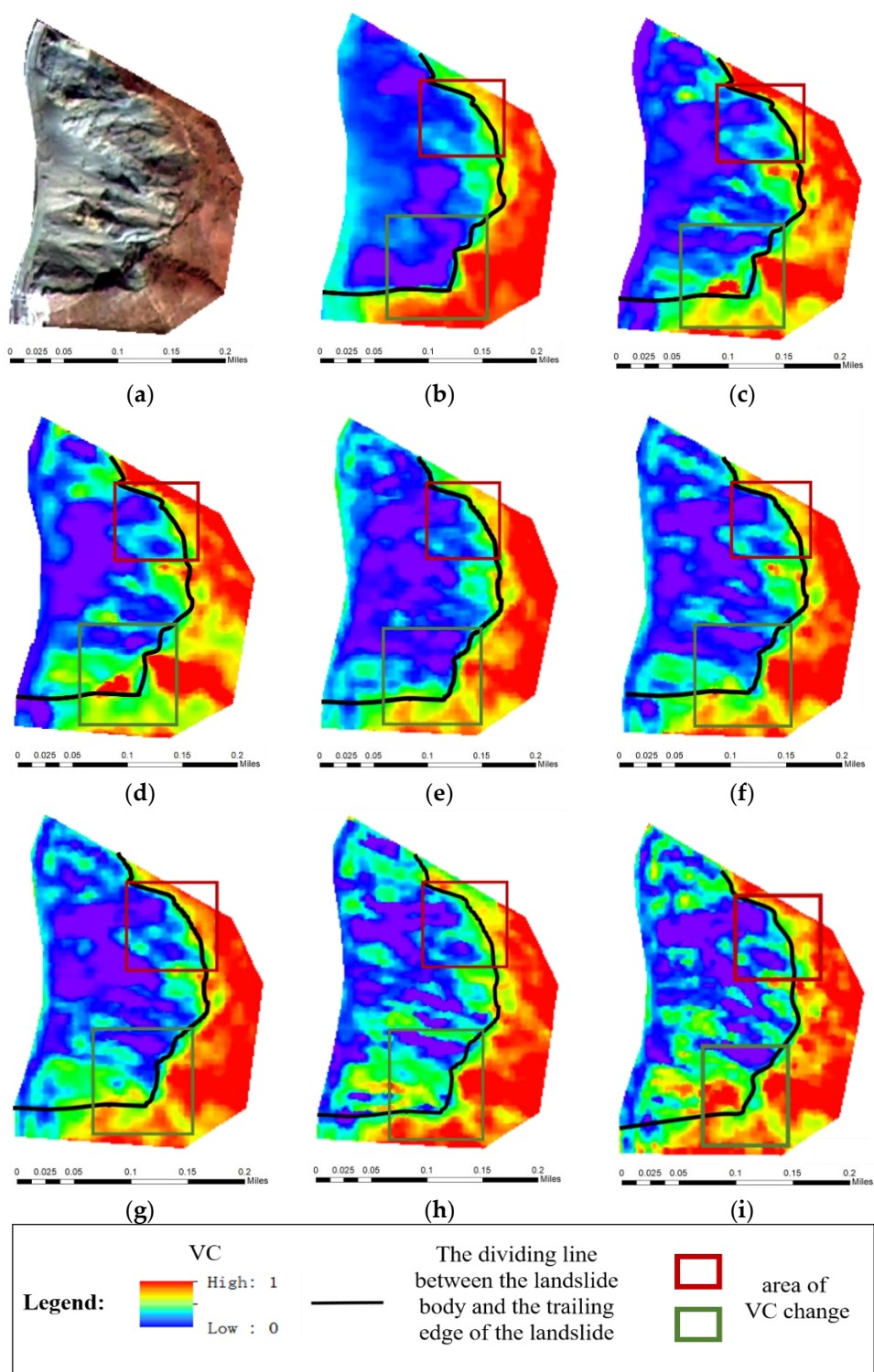

**Figure 5.** Pseudo-color VC maps in the upper of the Jizong Shed-Tunnel landslide. (**a**) Optical GF-1 fused true color image; (**b**) VC Map in 2013; (**c**) VC Map in 2014; (**d**) VC Map in 2015; (**e**) VC Map in 2016; (**f**) VC Map in 2017; (**g**) VC Map in 2018; (**h**) VC Map in 2019; (**i**) VC Map in 2020.

In the red box at the upper right corner, the bare land at the back edge of the landslide has a trend of gradual upward development and the VC is decreasing from 2013 to 2015, which corresponds to the known landslide in 2015. It provides a certain basis for the method of monitoring the Jizong Shed-Tunnel landslide creep with abnormal vegetation information. After 2016, the red area with high VC on the trailing edge of the landslide in

the red box gradually moves backward, and the original red area changes into yellow and green, which shows that the VC decreased.

In the green box at the lower right corner, from 2013 to 2015, the area next to the high vegetation red area protruding from the trailing edge of the landslide changes from red to yellow and green, which is consistent with the trend of the landslide in 2015. In 2016, the areas on both sides are slightly eased. After 2016, the protruded part on the trailing edge of the landslide in the green box gradually shrinks, and the bare land part gradually moves upward. The area under the green box also changes from red to yellow and green from 2016 to 2020, which expresses the decreased VC.

The above research and analysis are mainly based on the visual interpretation of the vegetation cover classification map of time series for qualitative analysis. In order to more accurately analyze the VC change and monitor the vegetation growth in the area of Jizong Shed-Tunnel landslide, the quantitative analysis and discussion are needed. For the whole area in Figure 5, we use the classification grade [52] to divide the VC and calculate the number and percentage of corresponding pixels for quantitative analysis. In order to correspond to the actual situation of the study area and reduce the influence of the shadow part of the landslide body on the result analysis, the low and medium VC is set $VC \leq 0.85$ (VC above 0.85 is the red area in the subplots of Figure 5). The statistics of the number of pixels and the cumulative percentage of low and medium VC are shown in Table 4. At the same time, the curve is made as shown in Figure 6.

**Table 4.** Number and percentage of pixels with low and medium VC.

| Image Time | $F_v \leq 0.85$ Pixels Number | $F_v \leq 0.85$ Pixels Percentage |
|---|---|---|
| 5 November 2013 | 21,103 | 76.39% |
| 8 November 2014 | 22,317 | 80.78% |
| 16 November 2015 | 23,525 | 85.16% |
| 22 December 2016 | 22,506 | 81.47% |
| 11 November 2017 | 22,138 | 80.14% |
| 19 November 2018 | 22,280 | 80.65% |
| 24 November 2019 | 22,522 | 81.53% |
| 30 November 2020 | 22,547 | 81.62% |

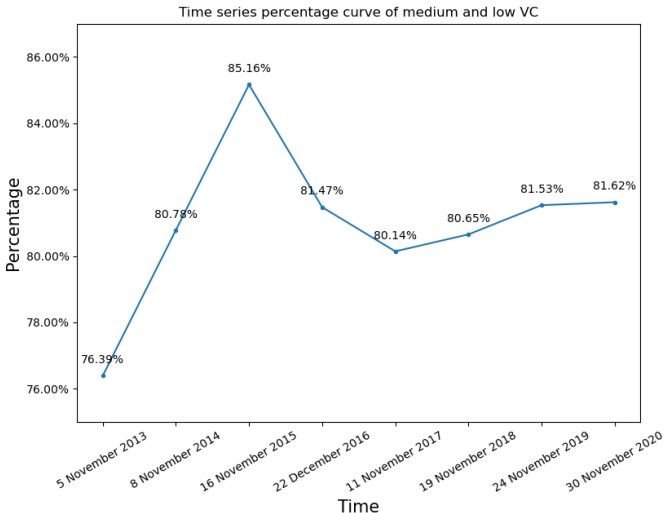

**Figure 6.** Time series change curve of the pixel number percentage with medium and low VC values.

According to the statistical results in Table 4 and Figure 6, the number and percentage of low and medium pixels gradually increased between 2013 and 2015, indicating that the vegetation growth situation gradually deteriorated, which is consistent with the above analysis results based on the VC classification map. With the time approaching the landslide in 2015, the vegetation growth became worse, with the maximum increasing percentage,

8.77%, of the low and medium VC, and the range of abnormal vegetation information expands. From 2015 to 2017, the number of the abnormal vegetation pixels decreased, which indicates that the vegetation had a certain recovery. After 2017, the pixels with low and medium coverage increased slowly, which is consistent with the conclusion that the overall VC change a little, but change in small areas.

According to the statistical results in Table 4 and Figure 6, the number and percentage of low and medium pixels gradually increased between 2013 and 2015, indicating that the vegetation growth situation gradually deteriorated, which is consistent with the above analysis results based on the VC classification map. With the time approaching the landslide in 2015, the vegetation growth becomes worse, and the range of abnormal vegetation information expands. From 2015 to 2017, the number of the abnormal vegetation pixels decreases, which indicates that the vegetation has a certain recovery. After 2017, the pixels with low and medium coverage increase slowly, which is consistent with the conclusion that the overall VC change a little, but change in small areas.

### 4.3. Surface Deformation Extraction Results Based on SBAS-InSAR

This paper uses the ENVI SARscape tool to process Sentinel-1 A data to obtain the average deformation rate of the ground highcoherence points and the cumulative surface deformation along the line of sight (LOS) of the satellite in the study area. However, because these points are scattered and only have latitudes and longitudes, it is inconvenient for visual interpretation and impossible to clearly observe and verify the specific surface deformation of the study area. Therefore, the average deformation rate map obtained by the SBAS-InSAR technology is superimposed with the GF-1 image. In order to better analyze the surface subsidence, only the coherent points (with the threshold 0.2 of the coherent points) of subsidence along the LOS direction are displayed, as shown in Figure 7.

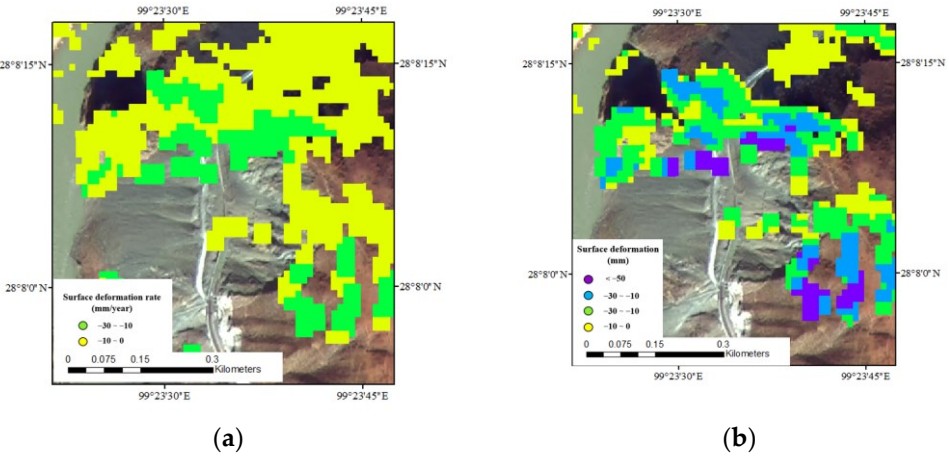

(**a**)                                    (**b**)

**Figure 7.** The surface deformation extraction result map of the study area (LOS direction). (**a**) The surface subsidence rate map; (**b**) The surface cumulative deformation map.

We can find that the settlement rate of the Jizong Shed-Tunnel landslide along LOS direction is mostly between 0 mm/year and 30 mm/year from Figure 7a. According to the sliding speed threshold in Table 5, that is also the annual average surface deformation rate, this landslide is identified as the very slow type or the slightly slow type [53], so we find that the Jizong Shed-Tunnel landslide is in the landslide creep stage after a large slide in 2015. From Figure 7b, it can also be found that the Jizong Shed-Tunnel landslide has obvious settlements as a whole, reaching more than 50 mm in some places, which also proves that the landslide is sliding.

**Table 5.** Classification of landslide type according to the sliding speed [53].

| Speed Grade | Landslide Type | Sliding Speed Threshold | Destructive Force Description |
|---|---|---|---|
| 1 | Very slow | <0.016 m/year | No damage will occur to buildings that have been protected in advance. |
| 2 | Slightly slow | 0.016 m/year~1.6 m/year | Some permanent buildings are not damaged;even if the building cracks due to sliding, it is repairable. |
| 3 | Slow speed | 1.6 m/year~13 m/month | If the slip time is short and the movement of the edge of the landslide is distributed over a wide area, the road and fixed structures can be preserved after several major repairs. |
| 4 | Medium speed | 13 m/month~1.8 m/h | Fixed buildings at a certain distance from the foot of the landslide can't be damaged; the buildings located on the upper part of the sliding body are extremely damaged. |
| 5 | Fast speed | 1.8 m/h~3 m/min | It has time for escape and evacuation; houses, property and equipment are damaged by landslide. |
| 6 | High fast | 3 m/min~5 m/s | The destructive power of the disaster is large, and due to its high speed, it is impossible to transfer all personnel, resulting in some casualties. |
| 7 | Super fast | >5 m/s | The destructive force is huge, the surface buildings are completely destroyed, and the impact or disintegration of the sliding body causes huge casualties. |

From the perspective of the spatial subdivision of landslides, the trailing edge of the landslide is the main deformation area. Not only are most of the coherent points located at the trailing edge of the landslide, but also the settlement rate and cumulative deformation of the trailing edge of the landslide are relatively large. In addition, the upper part of the landslide body has partial deformation, while the lower part of the landslide has no obvious deformation information, which is in line with the movement pattern of the landslide creep stage.

## 5. Discussion

Refarding the spatio-temporal analysis of surface deformation in abnormal vegetation areas, this paper superimposes the vegetation anomaly area information on the optical GF-1 image and compares it with the average land subsidence rate obtained by using the SBAS-InSAR technology in Section 4.3, as shown in Figure 8.

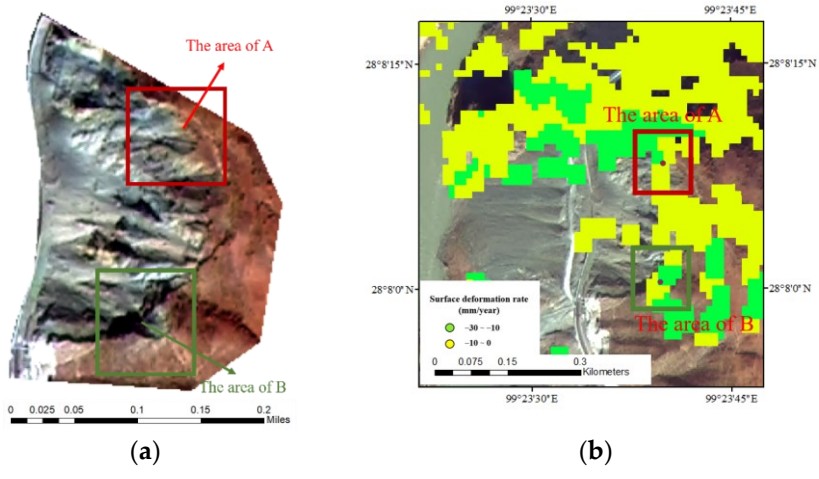

(**a**)  (**b**)

**Figure 8.** Location maps of the abnormal vegetation area. (**a**) The location on optical GF-1 fused true color image; (**b**) The location on the surface subsidence rate map.

According to Figure 8, this paper superimposes the abnormal vegetation A and B areas extracted from the GF-1 image to the surface subsidence rate map and finds that these two areas are located in the subsidence area of the coherent point in the surface subsidence rate map. At the same time, the subsidence range of these areas is 10 mm/year to 30 mm/year, which belongs to the landslide creep stage, reflecting the consistency between the monitoring of landslides through the abnormal vegetation information in optical images and the monitoring of landslides using the InSAR method.

Similar to the time series analysis of the abnormal vegetation information area, we also analyze the time series deformation of the A and B areas. Since the cumulative surface deformation obtained in SBAS-InSAR is displayed by points, this paper selects the center point of the vegetation anomaly area to approximate the cumulative surface deformation of the two areas. Moreover, we extract the low and medium VC in A and B areas for the quantitative analysis and make a specific comparison analysis, as shown in Figure 9.

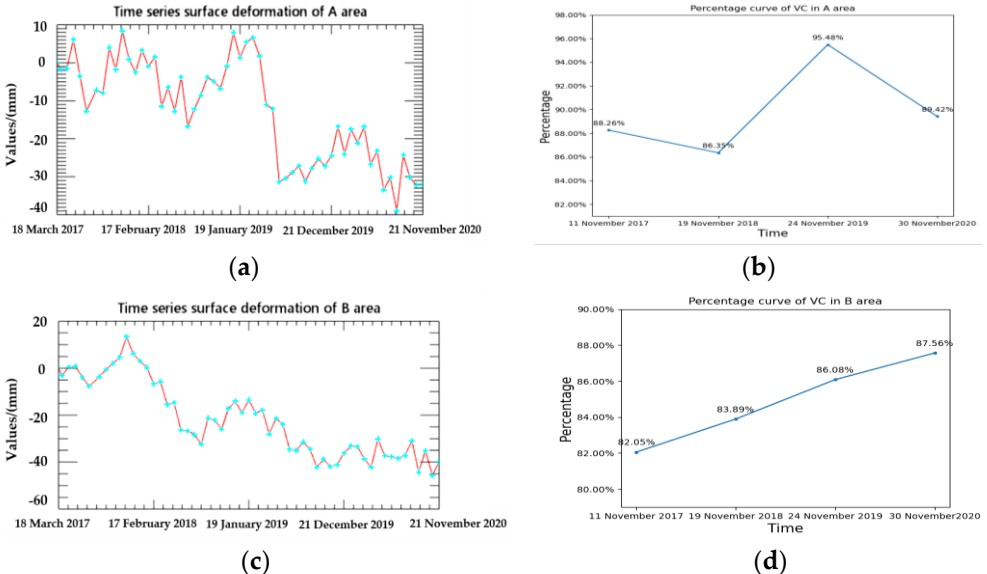

**Figure 9.** Time series diagrams of the abnormal vegetation information and the surface deformation. (**a**) Time series surface deformation of A area; (**b**) Time series of VC changes in A area; (**c**) Time series surface deformation of B area; (**d**) Time series of VC changes in B area.

According to Figure 9a, the overall deformation of the center point of A area is decreasing, and the maximum accumulated settlement is 40 mm. The time series deformation in this area rises slightly from June 2017 to March 2019, and the maximum amount of uplift is 9 mm, which may be caused by the surface movement of the surrounding area of the center point squeezing each other. Although the area has undergone certain deformation during this time period, it is still in a relatively stable state. In the percentage change curve of VC in Figure 9b, it is also found that the decrease in the percentage of low and medium VC from 2017 to 2018 indicates that the vegetation has recovered to a certain extent. This indicates that the surface deformation trend is consistent with the change trend of the VC. From January 2019 to November 2019, the overall time series surface deformation has a sudden downward trend. It seems that this area is unstable. After 2020, the time series deformation is not large, and the cumulative deformation is 7 mm, indicating that the area is still creeping. Similarly, in the percentage change curve of VC in Figure 9b, it is also found that the percentage of low and medium VC in 2019 has a rapid upward trend. It is basically consistent with the time series deformation trend of the ground surface. In 2020, the VC is slightly restored compared to 2019 but the overall trend is still declining.

According to Figure 9c, the maximum cumulative settlement at the center of B area is 45 mm. Except for a certain uplift from March to November 2017, the cumulative deformation of this area shows a downward trend, indicating that the area has been

creeping and deforming. From the percentage change curve of VC in Figure 9d, it can also be seen that the percentage of low and medium VC increased from 2017 to 2020, indicating a decline in VC. The overall trend is the same as the change in the surface deformation.

In order to more accurately analyze the correlation between vegetation anomaly information and surface deformation, this paper evaluates the accuracy of the two methods through the correlation and linear regression. In the previous comparison and analysis of the curves of the two methods, we find that the forms of surface activities in A area are relatively changeable, and B area is always in a creeping state and the form of the surface activity in B area is relatively stable. Moreover, since our study area is often covered by clouds, the optical images that we obtain most suitable for our conditions are all in autumn, which may cause the vegetation change to be less obvious. Therefore, we select B area as the typical analysis. Simultaneously, since the time of the used optical and SAR image is very difficult to be completely matched, and the time of the last SAR image in 2020 is earlier than that of the optical image, in order to maintain the time consistency, we select the SAR image time (20 October 2017, 8 November 2018, 3 November 2019, 21 November 2020) earlier than but closest to the optical image time (11 November 2017, 19 November 2018, 24 November 2019, 30 November 2020) for accuracy analysis, as shown in Figure 10. The x coordinate is the statistical average of the surface deformation. The y coordinate is the abnormal vegetation information, that is the percentage of the medium and low VC.

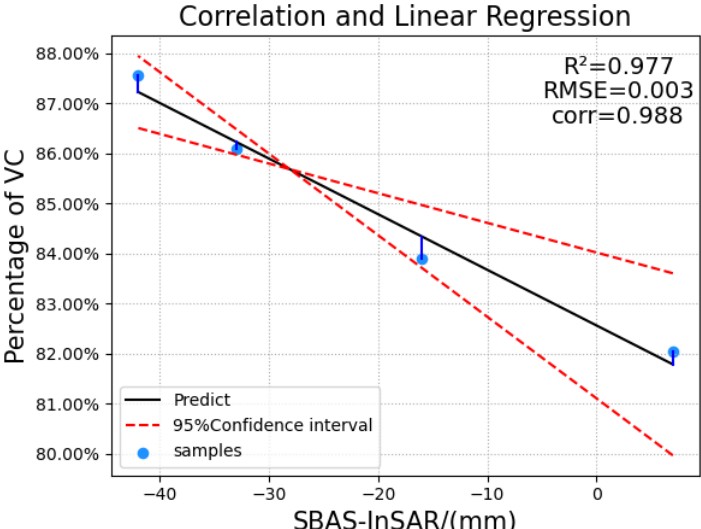

**Figure 10.** The correlation and regression analysis of the abnormal vegetation information and the surface deformation.

According to Figure 10, the correlation coefficient between the cumulative surface deformation and VC is 0.988, and the $R^2$ of the linear regression model is 0.977, indicating that the two methods have a significant linear correlation. When the deformation of the land surface subsidence increases, the pixels with medium and low VC also gradually increase, which provides reliable support for monitoring the Jizong Shed-Tunnel landslide using abnormal vegetation information. Simultaneously, the root mean squared error (RMSE) values of the error analysis and the upper and lower limits of the 95% confidence intervals are both small. This further proves the reliability of the accuracy analysis in B area.

Based on the comparative analysis and accuracy assessment of the cumulative time series variables and the changes in vegetation anomalies in these two areas, it was found that the vegetation anomalies are similar to the surface deformation variables, which proves that the use of abnormal vegetation information to monitor the Jizong Shed-Tunnel landslide has a certain degree of correctness and reliability.

## 6. Conclusions

In order to make up for the deficiencies of traditional GPS and InSAR technologies in monitoring landslides with large ups and downs, inaccessible by manpower, and lush with vegetation, based on the correlation between landslide creep and vegetation abnormality, this paper proposes a method to indirectly monitor the deformation characteristics of landslides by extracting the abnormal vegetation information from optical remote sensing images. We use the GF-1 optical data from 2013 to 2020 to monitor the vegetation anomaly information of the Jizong Shed-Tunnel landslide and use the SBAS-InSAR technology to extract the surface deformation information of the study area from 2017 to 2020. Then, we compare and analyze them. The results are as follows:

(1) This paper calculates the GNDVI index based on GF-1 time series data, and finally, obtains the vegetation coverage information of each scene. Through the multi-temporal qualitative and quantitative analysis of the extracted vegetation anomaly information, the VC decreased from 2013 to 2015. In reality, the landslide did occur in the study area in 2015, indicating that the early creep stage of landslides brings about a decrease in the VC. This verifies that the method of using vegetation anomaly information to monitor the Jizong Shed-Tunnel landslide is feasible. At the same time, it was discovered that there were two areas on the trailing edge of the landslide showing a downward trend in VC after 2017.

(2) Through the SBAS-InSAR technology based on the Sentinel-1 data, the main deformation area is located at the rear edge of the landslide, and the surface subsidence rate ranges from 0 mm/year to 30 mm/year, indicating that the Jizong Shed-Tunnel landslide is in a slow creep stage.

(3) After superimposing the abnormal vegetation area in the optical data with the surface deformation information in the radar data and performing time series analysis and accuracy assessment, it is found that the vegetation abnormality and the change trend of the surface deformation are basically consistent. When the surface deformation of the landslide decreases, the VC also shows a downward trend. When the deformation accelerates, the change in VC also intensifies. Even when the decline in the deformation is not large, the vegetation growth status can reflect these changes, which indicates the effectiveness and reliability of using vegetation abnormalities to monitor the Jizong Shed-Tunnel landslide, and the results of the two methods are similar. This method can provide new ways and ideas for the high-mountain landslide monitoring in southwestern China and can make up for some of the shortcomings of existing landslide monitoring methods.

Nevertheless, not all landslides at the creeping stage show obvious characteristic changes on surface vegetation, but some do exist. So, the landslide monitoring method in this paper is suitable for landslides with vegetation or vegetation change. Since the vegetation information needs to select the appropriate optical image with similar imaging time in each year, and the optical image is easily occluded by clouds, this method has some limitations in areas with cloud coverage. In addition, the impact of landslides on vegetation is a complex process, and this method has high monitoring accuracy for landslides that are in the creeping stage for a long time. In the future, we will also explore accurate pixel distinction models to improve the accuracy of vegetation information extraction.

**Author Contributions:** Q.G. proposed the research concept, designed the framework of this research, participated in data collection, reviewed and revised the manuscript. L.T. processed the GF-1 and Sentinel-1 data, analyzed data and wrote the manuscript. H.W. reviewed the manuscript and revised the figure. All authors have read and agreed to the published version of the manuscript.

**Funding:** This research was funded by the Leading Foundation on Frontier Sciences and Disruptive Technology Research of the Aerospace Information Research Institute, Chinese Academy of Sciences, grant number E0Z218010F.

**Data Availability Statement:** Thanks to the global availability of free and open Sentinel-1 SAR data with Europe Space Agency (ESA), the SAR data be accessible in https://sentinel.esa.int/web/sentinel/missions/sentinel-1 accessed on 20 March 2020; meanwhile, we are very grateful for the GF-1 data provided by China Centre for Resources Satellite Data and Application. Finally, we are also very grateful for SRTM DEM of 30 m resolution provided by United States Geological Survey (USGS); the DEM data be accessible in http://earthexplorer.usgs.gov/.

**Acknowledgments:** We are very grateful to the reviewers who significantly contributed to the improvement of this paper.

**Conflicts of Interest:** The authors declare no conflict of interest.

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
