# Peer review of "A Monitoring Method Based on Vegetation Abnormal Information Applied to the Case of Jizong Shed-Tunnel Landslide"

_remotesensing, doi:10.3390/rs14225640_

Round 1
Reviewer 1 Report
1. Line 10-11: I do not understand what do authors want to tell about the traditional method or their disadvantage? I think either authors present the traditional methods and their disadvantages or skip this sentence and directly stat about the objective or aim.
2. I do not find any quantification in the abstract section.
3. I can get overall idea about the work from Introduction. I think it is nicely written. Although, I think author should present more literature on vegetation abnormality after lines 79-80.
4. I think study area map needs to be revised. Need one national scale map, then authors can use regional scale map. Also, authors can present some details of the destruction of landslide, not mandatory.
5. The methods section is clear. I understand theme. However, my suggestion is that if the VC has low what does it indicates and vice versa. Please add this in the method section. Also, a flow chart would be advantage. Also, the methods lack the description of the evaluation process of pan-sharpening methods.
6. One of my critical assessment is the research needs a field based verification and accuracy assessment or not. Please write this in the method section.
7. Result is clear. Captions of figure need to be revised.
8. Discussion section does not have any discussion with other results; rather it has the extended result and analysis part.
9. Conclusion section is pretty large. Please make it concise and brief.
Author Response
We have answered the concerns of the reviewers in detail, point by point. Thank you very much indeed for the reviewers’ advice and the possibility to improve the manuscript.
Please see the attachment.

Reviewer 2 Report
I had read your manuscript carefully and found that your manuscript proposed a method for landslide monitoring based on vegetation abnormal information. And the results are comparatively analyzed with SBAS-InSAR results. The idea behind this work is commendable and I’d happy to see different methodologies are developed for landslide studies. However, considering the high quality of this journal, I think the manuscript should be improved as the following comments. Besides, the English language of the manuscript should also be improved.
Major comments:
1. Landslide monitoring is a major topic for geo-scientists, it includes deformation monitoring, pore pressure monitoring, stress monitoring and etc. However this manuscript just mentioned landslide monitoring, leading to the topic too general. So, I think the object of your monitoring method is unclear. (As your manuscript described, I guess maybe your method is a mainly aim to landslide deformation characteristics monitoring.)
2. About the principle of using vegetation abnormal information. The authors described ’Landslide creeping is ...vegetation on the landslide’. However, as I know, not all the landslide at the creeping stage owns obvious characteristics changes on surface vegetation but some of do exists (like Baige landslide and Su village landslide). So, the scope of the landslide monitoring when using vegetation abnormal information should be given. Besides, in the introduction section(line 71-84), most of your citations are about post-landslide identification, but not landslide monitoring implemented with optical images. I think the authors should focus on monitoring but not identification.
Ouyang C, An H, Zhou S, et al (2019) Insights from the failure and dynamic characteristics of two sequential landslides at Baige village along the Jinsha River, China. Landslides. https://doi.org/10.1007/s10346-019-01177-9
Ouyang C, Zhao W, An H, et al (2019) Early identification and dynamic processes of ridge-top rockslides: implications from the Su Village landslide in Suichang County, Zhejiang Province, China. Landslides 16:799–813. https://doi.org/10.1007/s10346-018-01128-w
More minor issues
1. Line 40. ‘Many scholars have done a lot of works on the landslide identification.’ Did you mean monitoring? Monitoring and identification are different.
2. Line 41 to 56. GPS is one of GNSS system. Please use those two words correctly.
3. Line 67 to 70. How did your solve this limitation for you study. Because your study had used InSAR technique and your study area also location in the Yunnan province.
4. Line 85 to 88. All the citation papers are researches of the authors. Are there other researches can support your opinions?
5. Line 125 Figure 1. improve the quality of the figures.
6. Line 156. ‘with a time interval of 24 days’. This is incorrect. the images taken on 2017-03-18 and 2017-03-30 just own 12 days interval. Please check the dataset or the description. Beside the orbit path and fly direction of S1 should also be given.
7. Line 185. “de-scribe”. Typing error.
8. Line 199. “HHerein”. Typing error.
9. Line 239 to 241. ‘Since the PS ... artificial targets.’. This statement is incorrect. The pixels with stable phase are selected by the amplitude dispersion index in the PS candidates selection.
10. Line 241 to 242. This is not the reason why SBAS technique should be used.
11. Subsection 4.1. This would be a supplementary information. Because this comparison sees have no direct relationship with your study. Please reconsidering this section.
12. Line 366. ‘On the left side of the black line’. Is the black line means the landslide boundary ? How do you define the edge of the line.
13. Line 397 to 399. Which area did you statistic?
14. Line 430. Give the threshold of the coherent points.
15. Line 433 to 435. This sentences confused me. please improve this sentence.
16. Line 440. Is there has relationship between coherent points and deformation? As I known, present studies only shows the large deformation rate could lead to incoherent of images, but there still no researches show deformation can lead to the increase of coherence of image.
17. Figure 7. The SBAS results sees has low quality. High quality InSAR result usually have not or few points located in the river range. So, please comparing your InSAR results with other monitoring data (GPS, leveling monitoring) or give the regional deformation map to show the quality of your results. (If the InSAR results own high quality, the regional deformation should be nearly zero.). Besides, give the reference point, if you had used.
InSAR-derived digital elevation models for terrain change analysis of earthquake-triggered flow-like landslides based on ALOS/PALSAR imagery. Environmental Earth Sciences, 2015, 73(11): 7661-7668
The Maoxian landslide as seen from space: detecting precursors of failure with Sentinel-1 data. Landslides 15:123–133. https://doi.org/10.1007/s10346-017-0915-7
An InSAR and depth-integrated coupled model for potential landslide hazard assessment. Acta Geotech. https://doi.org/10.1007/s11440-021-01429-w
Integration of Sentinel-1 and ALOS/PALSAR-2 SAR datasets for mapping active landslides along the Jinsha River corridor, China. Engineering Geology 284:106033. https://doi.org/10.1016/j.enggeo.2021.106033
18. Figure 8. In the VC analysis, it is conclude that the VC can reflect the landslide surface change. However, in the Figure 8, it can be see the high deformation at the right-down area of the image but hardly to see the VC changes in the counterpart area of Fig. 5. How to explain this?
Author Response

(The authors gave the same response as above.)

Reviewer 3 Report
The paper is quite well arranged and the description of methods and results are clear enough. As the topic is quite extensive and well investigated by the scientific community, some additional references should be included to clarify the main subject of the paper (monitoring as central focus) and the positioning of this research respect the investigation ongoing in similar topics. In addition, and taking into account landslide monitoring systems/methods are subject of several investigation I suggest to specify the aspects the authors are considering as innovative and relevant of their proposed method and the limit of the it as well as the possible synergic aspects with others methods. Concerning the specific comments referring to line numbers, tables or figures, these have been already reported in the report submitted. Anyway, for your continence they are reported below: - line 254 - 280 the numbered list of steps is confused with the list of formulas. Maybe you can use letters and numbers for the two lists to make easier the understanding. – - table 4 It could be used "Image time" instead of time in the first column as done on Table 1 and 2. An overall review of English could be valuable to make the paper easier to understand for the readers.Author Response
We have answered the concerns of the reviewers in detail, point by point. Thank you very much indeed for the reviewers’ advice and the possibility to improve the manuscript.
Please see the attachment.
